# A Review on Digestive System of *Rhynchophorus ferrugineus* as Potential Target to Develop Control Strategies

**DOI:** 10.3390/insects14060506

**Published:** 2023-05-31

**Authors:** Ahmad-Faris Seman-Kamarulzaman, Faizatul Atikah Pariamiskal, Amiratul Nabihah Azidi, Maizom Hassan

**Affiliations:** 1Institute of Systems Biology, Universiti Kebangsaan Malaysia, Bangi 43600, Selangor, Malaysia; afkamarulzaman@uitm.edu.my (A.-F.S.-K.); p115980@siswa.ukm.edu.my (F.A.P.);; 2Faculty of Applied Sciences, Universiti Teknologi MARA Pahang, Bandar Tun Abdul Razak Jengka 26400, Pahang, Malaysia

**Keywords:** red palm weevil, gut, potential approach, pest management, omics

## Abstract

**Simple Summary:**

The red palm weevil poses a significant threat to palm species, resulting in substantial economic losses. While multiple methods have been developed to control its infestations, there is an urgent need for eco-friendly insecticides that selectively target its critical systems or pathways. One such target is its digestive system, which is essential for its survival. This review highlights the potential of using the digestive system of the red palm weevil to manage its infestations. Proteomic and transcriptomic data analyses on the weevils have provided a better understanding of the protein and gene compositions in its digestive system. With technological advancements, a more comprehensive approach can be taken to explore the opportunities in manipulating the data on the digestive system of red palm weevil, leading to improved management methods.

**Abstract:**

*Rhynchophorus ferrugineus,* commonly known as red palm weevil (RPW), is a high-risk insect pest that has become a threat to many important palm species. There are several dominant factors that lead to the successful infestation of RPW, including its stealthy lifestyle, highly chitinized mouthpart, and high fecundity rate. Due to that, millions of dollars of losses have been suffered by many countries invaded by RPW. Several methods have been designed to control its invasion, including the usage of insecticides, but many cause resistance and environmental pollution. Therefore, an environmentally friendly insecticide that targets specific systems or pathways in RPW is urgently needed. One of the potential targets is the digestive system of RPW, as it is the major interface between the insect and its plant host. The related knowledge of RPW’s digestive system, such as the anatomy, microflora, transcriptomic analysis, and proteomic analysis, is important to understand its effects on RPW’s survival. Several data from different omics regarding the digestive systems of RPW have been published in separate reports. Some of the potential targets have been reported to be inhibited by certain potential insecticides, while other targets have not yet been tested with any inhibitors. Hence, this review may lead to a better understanding on managing infestations of RPW using the system biology approach for its digestive system.

## 1. Introduction

*Rhynchophorus ferrugineus* (Coleoptera: *Curculionidae*), commonly known as the red palm weevil (RPW) (Olivier 1790), is a Coleopteran insect that has been classified as a significantly serious pest on the A2 list (pests are locally present), according to the European and Mediterranean Plant Protection Organization (EPPO) [1]. RPW has been reported to invade various economically important palm species on almost all continents, except Antarctica [2,3]. Their invasion has cost 30% of the world’s date palm (*Phoenix dactylifera*) production loss in the Middle East region, with an estimation of 130 million USD annually [4]. In addition, the infestation of this pest has caused fatal damage to nearly 200,000 young coconut palms (*Cocos nucifera* L.) in Sri Lanka, resulting in a financial loss of about 1.8 million USD in 2005 [5]. In Malaysia, RPW infestations have rapidly spread in coconut plantation areas in Terengganu, from 58 localities in 2007 [6] to 858 localities in 2011. Their infestation was further reported to have spread to three other states in the northern region of Peninsular Malaysia in 2016.

RPW is a sexually dimorphic [7] and holometabolous insect, as it has four developmental stages that consist of egg, larvae, pupae and adult (Figure 1) [8]. It takes about 45 to 298 days to complete the whole lifecycle, depending on their diet [9]. The female RPW has a high fecundity rate, producing around 180–396 eggs throughout its lifespan [10].

RPW has been reported to be most destructive in its larval stage [11]. The larvae feed on the trunk, creating empty cavities inside and leading to the death of the tree. It is difficult to identify infested palm trees in palm plantations due to the lack of obvious symptoms of infestation, especially at the early stage of infestation [3,12]. The remarkable adaptation of the concealed RPW’s lifecycle also makes it hard to remove the pest from the palm tree besides dissection [7]. The damage from the RPW infestation is fatal, as the physical symptoms are only visible after the tree has been severely damaged, thus destroying the tree beyond saving before the pest is detected [1].

The current strategies reported for the prevention and control of RPW are based on a natural enemy as a biocontrol agent, food-baited pheromone traps and insecticides [6]. Out of the various choices, broad-spectrum insecticides are currently the main strategy for RPW control [13,14,15,16]. However, the usage of commercial broad-spectrum insecticides, such as phosphine [17] and ethion [18], has been proven ineffective, as RPW develops a resistance towards them [16]. Moreover, they pollute the environment, and they kill nontargeted beneficial insects [4]. Accordingly, the insufficient current management [19] leads to the need for in-depth studies for RPW control, such as using molecular approaches that target its specific biochemical system. One of the potential systems is the digestive system of RPW, as it is one of the ways to control its infestation, but little is known about it [13,20].

Hence, this article aimed to review the available information regarding RPW’s digestive system. The knowledge related to RPW’s digestive system, such as its anatomy, microflora, transcriptomic analysis, and proteomic analysis, especially factors that may affect its feeding behavior, could be the first step in exploring the possibility of new treatments to manage RPW infestations in palm tree species [12].

## 2. Anatomy of RPW’s Digestive System

Within the body system of an insect, the most crucial one in the insect’s life is the digestive tract system. The digestive tract is basically responsible for supplying the essential nutrients for survival and growth, as well as carrying out daily activities [15]. RPW feeds on the fibrous part of the host plant using its mouthpart. One of the causes that leads to a successful infestation of RPW is its highly chitinous mouthpart [21]. As a phytophagous insect that feeds on palm species, RPW mouthparts consist of strong mandibles with a bite-chew mechanism. The characteristics of the mouthpart enable RPW to drill into the stem of the host plant. This is different from the mouthparts of beneficial insects, such as pollinators (Hymenoptera) that utilize fluid-feeding mechanism using a proboscis [22].

When the damage due to infestation worsens, a chewing sound can be heard when one’s ear is close to the palm trunk [23]. In addition to the RPW’s strong mouthpart, this economically important pest also possesses an alimentary gut system that can digest the high-fiber food that it ingests. The gut can be divided into three different parts (Figure 2): foregut, midgut and hindgut [13,15,24]. Mainly, the foregut is used to eat the food, and the hindgut is for the absorption of water. The secretion of enzymes and digestion of food occur in the midgut [13].

The digestive organ of RPW is reported as slightly different between larvae and adults due to the differences in their feeding modes [13]. According to Harris et al. [15], the larvae have larger organs, as they consume more food and need more energy for periodic molting process. Any hindrance to the digestive system of the larvae may result in the death of the larvae due to nutrient deficiency and insufficient energy. Hence, the gut in larvae can become a great target to control RPW infestations.

## 3. Gut Bacteria of RPW

Other than the anatomy of the gut, the content of the gut is also similarly important, especially the microflora which greatly helps to enhance food digestion in RPW. It was reported that insect gut microbiota was generally dominated by *Proteobacteria* and *Bacteroidetes* [25,26,27]. One study revealed that the gut bacteria present in RPW exist in six abundant groups of bacteria with different phylogenetic group ranks: *Enterobacteriaceae*, *Leminorella grimontii*, *Entomoplasmatales*, *Erysipelothrix*, *Lactobacillus* and *Leuconostoc* [28]. However, this composition and population dynamics can vary due to environmental factors, including the diet of the insect [29].

The gut microbiota was reported to promote effects on the growth, development, mating and immunity of RPW larvae [30,31]. RPW’s gut was also reported to be equipped with bacteria that provided tyrosine, which is essential for the formation of the insect’s cuticle and hardening [32]. An infection of aposymbiotic RPW with *Serratia marcescens* and *Escherichia coli* showed a significantly faster rate of death compared to the symbiotic RPW. Therefore, the gut microbiota was proven to benefit RPW in its survival and immune defense [31]. In addition, the gut bacteria were reported to be able to manipulate the way in which RPW processed and utilized nutrients. It was shown that the body mass, protein content and glucose levels were higher in conventionally reared RPW compared to germ-free RPW [33].

To further investigate the microbiota present in the gut of beetles, especially RPW, various approaches involving metagenomic studies can be utilized [34]. Some of the approaches are 16S rRNA gene sequencing [35], shotgun metagenomics [36] and metatranscriptomics [37]. The usage of the 16S rRNA gene sequencing approach was reported to successfully identify seven potential gut bacteria to be targeted for the management of RPW. The gut bacteria identified were *Serratia enterica*, *Enterobacter cloacae*, *Raoultella* sp., *Klebsiella pneumonia*, *Klebsiella variicola*, *Klebsiella oxytoca and Citrobacter koseri*. Those seven bacteria were determined to have a great cellulose degradation ability. Disrupting the symbiosis between RPW and the bacteria may result in a significantly affected nutrition metabolism of the RPW [35].

The utilization of the shotgun metagenomics approach on RPW was reported to identify the two most abundant species, namely *Klebsiella pneumoniae* and *Lactococcus lactis* [36]. *K. pneumoniae* was also found in the spruce bark beetle *Dendroctonus micans* [38], and it has been suggested to be involved in cellulolytic digestion [39]. *L. lactis*, a type of lactic acid bacteria, plays a significant role in the digestion and fermentation of plant polymers in the gut of RPW to enhance the insect’s ability to obtain nutrition from its diet. Moreover, *L. lactis* carries out lactic acid fermentation, which converts carbohydrates into lactic acid or lactate. This helps to break down carbohydrates into a more usable form for the insect and maintains an acidic environment in the gut, promoting the growth of beneficial bacteria and inhibiting the growth of harmful bacteria [33]. Both bacteria were suggested to be potential candidates for further studies on gut indigestion in RPW. Other than that, a pathogenic virus called *Cotesia sesamiae* bracovirus (CsBV) was also discovered in RPW’s gut in the same study [36]. CsBV is a type of virus that belongs to the *Polydnaviridae* family. It is a symbiotic virus that is carried by the parasitic wasp *C. sesamiae* and is injected into the host insect when the wasp lays its eggs inside the host’s body [40]. Previously, the cytoplasmic polyhedrosis virus has been reported to affect RPW, resulting in malformed adults [41]. Hence, Jia et al. [36] suggested that the insect-infectious bracovirus is worth being further studied as a candidate pathogen for effective RPW biocontrol. In addition, host-borne bacteriophages were identified in the gut of RPW [36]. Several bacteriophages have been known to cause mortality in RPW larvae [42,43], and therefore, the identified bacteriophage could also infect a range of gut bacterial species and cause an imbalance in RPW’s gut microbiota population [36].

Meanwhile, the metatranscriptomic approach, which can identify actively expressed genes in complex microbial communities, has been used to study the gut microbiota of beetles but not yet in RPW [37]. The metatranscriptomic approach was used, as it is a powerful tool that can reveal the actively expressed genes in complex microbial communities [44]. The flea beetles, namely *Altica fragariae* and *A. viridicyanea*, feeding on *Duchesnea indica* and *Geranium nepalense*, respectively, were compared after being swapped between hosts, revealing differences in their microbial communities and enriched genes identified in the gut [37]. These genes were involved in breaking down the secondary metabolites produced by the host plant, and the microbial communities were found to support their function. The gut microbiota helped the beetle to feed on and adapt to their potentially toxic host plants, highlighting the link between the functions of these genes and the diversity of the microbial communities [37]. The link between the diversity of gut microbiota and gene functions was also shown in grasshoppers [45]. Therefore, applying the metatranscriptomic approach to RPW’s gut could reveal the potential gut microbiota that cooperates with RPW, contributing to understanding its adaptation to different host plants and potentially managing its survival [37]. The cooperation between the insect’s gut microbiota, which leads to a better metabolism for the host’s nutrition, has also been shown in sugarcane borer *Diatraea saccharalis* [46]. Zhang et al. suggested that the understanding in the host adaptation could be explored by the utilization of an artificial feeding system, incorporating in vitro cultured gut wall cells to replicate the insect gut environment. This system allows for the assessment of host nutrition–gut microbiota interactions [47]. In summary, exploring the dynamics of the microbiota diversity present in the gut of RPW using various approaches in the metagenomic analysis is crucial for developing effective biocontrol strategies to manage this pest [11]. It is a promising alternative method to control this pest by disrupting the interactions between RPW and its gut microbiota [33,35].

## 4. Protein Profile of RPW’s Gut

Another important component in the digestive system of RPW besides the microflora is the digestive enzyme. Digestive enzymes play an essential role for proper and efficient digestive system function (Table 1). The presence of certain digestive enzymes in the system enables RPW to digest the foods it consumes. A thorough study about digestive enzymes in their gut enabled the discovery of the relationship between the insect and its environment, indicating the degree of food preference. Therefore, identifying digestive enzymes through liquid chromatography-mass spectrometry (LC-MS) from the protein profile of the RPW gut reared on different diets, namely coconuts, oil palm and sago, can provide insights into the preference of the host plant for an RPW infestation. This approach is valuable for understanding RPW’s digestive system and its interaction with the environment [20]. Out of the three diets tested, coconuts were found to contain higher protein contents and were also the most preferred by RPW. The preference of RPW towards coconuts matched with the results of the protein profiles, with the major enzymes identified in the RPW’s gut involved in the digestion of proteins being trypsin and aminopeptidase. The study discovered that the digestive enzyme matched with the dietary ingredient and a higher preference of the ingredient in the food for the larvae to digest [20].

Other studies showed similar results matching digestive enzymes with food preferences. One study revealed that carbohydrate-metabolizing enzymes were more active in the gut of maize stem borer *Busseola fusca* [48] and fall armyworm *Spodoptera frugiperda* [49], as both insects had carbohydrates as their major food source. In addition, the protease enzyme has also been characterized in the gut of RPW [50] and proven to be important as a target after it was tested for its inhibition with several insecticides in a separate study [51].

Apart from protein-metabolizing enzymes, the most common hemicellulase enzyme [52], xylanase, was found in the gut of RPW [53]. Xylanase (EC: 3.2.1.8) is an enzyme that hydrolyzes xylan (Table 1), which is the major constituent of hemicelluloses of plant cell walls. In addition, the carbohydrate-active enzyme called glycosidase, which is responsible for hydrolyzing complex carbohydrates and polysaccharides of plant cells into smaller products [54], and their protein family, namely amylase, have been identified and characterized in the gut of RPW [50,55,56,57]. Glycosidase (EC: 3.2.1), also referred to as glycoside hydrolase, is an enzyme that catalyzes the degradation of cell wall polysaccharides [58]. Amylase (EC: 3.2.1.1), which also belongs to a glucoside hydrolase family, is an enzyme that catalyzes the D-(1,4)-glucan linkage in starch and related carbohydrates in plant cells [55]. The existence of these enzymes is consistent with the behavior of RPW, which consume plant tissues. The silencing of the digestive enzyme, such as amylase [59] in RPW and xylanase [60] in coffee berry borer *Hypothenemus hampei* through injection of its respective dsRNA, was reported to cause mortality of the larvae.

In addition, the high activity of antioxidant enzymes (Table 1), such as catalase [61], polyphenols oxidase and peroxidase, involved in insect defense mechanisms were also detected and analyzed in the gut of RPW [62]. Antioxidant enzymes in insects are enzymes that can balance potentially harmful reactive oxygen species (ROS). ROS are produced due to changes in the biotic or abiotic factors, as well as exposure to insecticides [63]. Catalase (EC: 1.11.1.6) and peroxidase (EC: 1.11.1.7) function to catalyze the conversion of potentially toxic hydrogen peroxide into oxygen molecules and water [64]. Polyphenol oxidase (EC: 1.14.18.1) is an enzyme that uses oxygen to catalyze the oxidation of a wide range of phenolic compounds resulting from several factors, such as exposure to insecticides and starvation [62]. The functions of the antioxidant enzymes found in the gut of RPW showed that each of the enzymes could be targeted to control the pest. The enzymes were listed as some of the key enzymes of great potential to be targeted and knocked down using the RNAi approach [65]. It was proven that, when catalase was knocked down in an RNAi-based experiment, significant mortality and growth inhibition of RPW’s larvae [66] and *Spodoptera litura* (Fabricius) larvae were recorded [67]. Moreover, the silencing of the catalase gene using the RNA interference approach in *Amblyomma maculatum* has resulted in a decrease in both egg mass and larval eclosion rates [68]. Furthermore, the knockdown of peroxidase [69] and polyphenol oxidase [70] in wheat aphid (*Sitobion avenae*) also led to a reduced survival rate and ecdysis index. Additionally, the RNAi-mediated silencing of salivary gland peroxidase in *Anopheles gambiae* caused a lower blood-feeding capacity [71], while the silencing of polyphenol oxidase in *Bombyx mori* caused incomplete pupation [72].

Most insects are highly dependent on the enzymes present in their gut for development and survival [73]. Therefore, analyzing the activity of the digestive and antioxidant enzymes in RPW’s gut can contribute to the development of new insect pest control by preventing the digestion and assimilation of nutrients in the insect. Furthermore, the knockdown and silencing of potential digestive and antioxidant enzymes using RNAi is a promising and powerful approach to control and manage insect pests such as RPW [65].

## 5. RPW’s Gut Transcriptome Analysis

RPW is a phytophagous pest that devours the wet woody trunk and very sugary sap of palm trees [74]. Similar to other herbivorous pests that feed on woody plants, these pests must detoxify the secondary metabolites, such as allelochemicals, produced by the plants [75]. This often leads to the development of metabolic adaptations that require various types of detoxifying enzymes. In some cases, the same enzymes may be responsible for both the pest’s adaptation to natural plant defenses and its resistance to insecticides [76]. These metabolic adaptations may be the result of preexisting detoxifying enzymes within the pest’s body, enzymes provided by microbial symbionts or enzymes acquired through horizontal gene transfer from fungi or bacteria [1].

There have been several reports stating the discovery of detoxifying genes (Table 2) in RPW’s gut. Cytochrome P450 (CYP450) and glutathione-S-transferase (GST) were reported to be highly expressed in the mid-gut of RPW [62,75,77,78]. The detoxification system of insects consists of three phases: biotransforming, metabolizing and excreting. CYP450 was grouped into phase I, while GST was grouped into phase II [79]. CYP450 is a heme-containing protein that aids in the detoxification of insecticides by catalyzing the oxidation of the insecticide, leading to an increase in its solubility. This increase in solubility facilitates the removal of the insecticide from the body of the insect, resulting in its detoxification [77]. CYP450 can also work in conjunction with GST to detoxify insecticides. After the initial oxidation of the insecticide by CYP450, glutathione is added by GST to the oxidized compound to further enhance its elimination from the insect’s body [1]. An RNAi experiment performed on CYP450 found in RPW showed an increasing insecticide susceptibility that led to RPW’s death after it was tested with cypermethrin [4].

In addition, the transcript of a laccase enzyme (EC: 1.10.3.2) was found to be expressed in the gut of RPW (Table 2). This enzyme is a cuticular protein responsible for cuticle hardening to protect insects from environmental stress, and it may oxidize toxic compounds ingested by insects [81]. The injection of dsRNA into *Tribolium castaneum* designed for an RNAi experiment resulted in the depletion of laccase transcripts and caused the weevils to fail to tan, producing soft-bodied weevils that subsequently died [82,83].

Furthermore, a genome-wide analysis was conducted on digestion-related genes from the transcriptome data of female and male adult RPW. The analysis revealed 70 glycoside hydrolase genes, 17 α-amylase genes, 13 carboxypeptidase genes and 1 chitin synthase gene. The existence of carboxypeptidase (EC: 3.4.16.2) is important for stem borers such as RPW to catalyze the digestion of protein in the trunks of palm trees (Table 2) [84]. The significance of carboxypeptidase was demonstrated when the larvae of *Cosmopolites sordidus*, a banana weevil, showed lower survival rates and displayed a significant reduction of growth when it was inhibited by a soybean protease inhibitor [85].

On the other hand, chitin synthase (EC: 2.4.1.16) catalyzes the polymerization of the chitin polymer [86], a major component of insect cuticles, including the chitin layer that lines the gut of RPW (Table 2). This layer provides RPW with protection from rough food particles and ingested pathogens [1]. The inhibition of chitin synthase in cotton boll weevils (*Anthonomus grandis*) using RNAi recombinant transgenic cotton resulted in malformed first and third instar larvae. The report suggested that the knockdown of chitin synthase led to the obstruction of nutrient uptake in the gut [87].

In another study, the expression profiling of RPW’s gut showed the identification of key enzymes in the digestion of plant cell walls (Table 2), namely cellulase (EC: 3.2.1.4), hemicellulase and pectinase (EC: 3.2.1.15) [54,88]. Those three enzymes function in catalyzing the degradation of major polysaccharides in the cell walls of most plant cells, including cellulose, hemicellulose and pectin [52]. As RPW is the major pest for many plant species by attacking and feeding on plant stems, the inhibition of these enzymes in RPW will lead to indigestion and starvation, eventually resulting in death [89].

Other than that, potentially targeted genes encoded for neuropeptide precursors and their receptors (Table 2) were found to be predominantly expressed in the gut of RPW. In insects, neuropeptides interact with their receptors and trigger signal transduction and physiological processes such as feeding and digestion [90]. A study found that an analogue insecticide targeting neuropeptides was applied to the insect pest *Myzus persicae*, resulting in a high mortality rate. This study also highlighted that the insecticide tested was not harmful towards beneficial insect *Chrysoperla carnea* [91].

Hence, the gut transcriptome analysis of RPW shows that it can offer valuable insights into cellular processes that occur in the gut. The identification of those genes and neuropeptides is necessary for the development of potential insecticides to effectively manage and prevent the invasion of RPW in palm trees [11,88,92].

## 6. *R. ferrugineus* Control Related to Its Digestive System

Having a comprehensive knowledge regarding the potential targets in the digestive system of RPW is helpful in developing and discovering new potential strategies in managing the RPW pest.

Although certain potential targets have been found to be responsive to insecticides or inhibitors [93], several other potential targets located in the gut of RPW have only been reported and not yet been specifically subjected to any insecticidal evaluation (Table 3). This creates a prospect for further research.

According to Zulkifli et al. [20], among the alternative methods in managing RPW infestations is targeting the potential proteins involved in nutrition or digestion. Identifying selective inhibitors of the digestive enzymes and potentially targeted proteins or genes in the RPW gut is important to turn off the enzymatic activity.

The discoveries of selective inhibitors, such as specific inhibitors targeting the trypsin and peptidase enzymes, which are crucial components of insecticides, can result in the disruption of digestion in the target insect. Consequently, this can impede the insect’s growth and eventually lead to the mortality of the pest weevil. This corresponds with the findings by Mohamed et al. [53], who emphasized the importance of understanding the substances that inhibit digestive enzymes, such as the xylanolytic enzyme, as it can contribute to the development of control strategies for RPW infestations. An example of selective inhibitors is proteases inhibitors (PIs), which function as antimetabolic proteins that inhibit the insect’s digestive activity. Orfali et al. [95] conducted research that demonstrated date kernel extract and *Calotropis latex* extract PIs were effective in inhibiting and reducing the activity of digestive protease enzymes in the gut of RPW by 39% and 18%, respectively, resulting in a decrease in the survival rate of the insect. This suggests that PIs obtained from plant extracts can serve as a valuable bioactive source for the development of biopesticides, essential for maintaining an ecological balance in the management of RPW insect pest infestations.

Another way of controlling RPW is by targeting the specific biosynthesis pathway involved in the digestive system of RPW. One of the potential target pathways for pest control in RPW is the chitin biosynthesis pathway [18]. As mentioned before, one of the transcripts found and suggested to be targeted is the chitin synthase (CHS2) gene, as it is involved in developing chitin in the gut lining of RPW. Chitin also builds up the well-developed mouthparts of adult and larvae RPW, which are strongly chitinized, enabling the pest to destroy the rough components of the palm vascular system [14]. It has been proven that chitin synthesis inhibitors such as chlorfluazuron, hexaflumuron and lufenuron showed efficiency against RPW larvae. Hence, the identification of chitin degradation-related transcripts from the digestive tract of RPW unfolds the RPW chitin degradation mechanism that might be manipulated for the development of targeted and specific future molecular insecticides [18].

## 7. Opportunity and Challenges of Targeting RPW’s Digestive System

Studying RPW’s digestive systems is crucial in agriculture, especially because of the detrimental impact of RPW as insect pests. With the advent of high-throughput analysis in metagenomics, proteomics and transcriptomics, the investigation of insects’ digestive systems has progressed to the systems’ biology level, encompassing the whole system rather than individual components. This approach offers insights into the dynamic nature of the digestive system and creates opportunities for further exploration [44].

The ability of phytophagous insects to survive is largely dependent on their capacity to effectively consume their respective host plant. As for RPW, its ability to consume and digest the host plant was reported to be aided by its insect-associated microbial communities in the gut [35,36]. Identifying and manipulating the microbial communities in the gut to manage the infestation of RPW is considered a novel management strategy [30]. However, the dynamic relationship between the symbiont’s microbe communities and RPW remains poorly understood. As the diversity of the microbes can vary due to several factors, such as different host plants [37], temperature and pH [36], the dynamics need to be further investigated [35].

Additionally, gaining knowledge on the digestive enzymes that RPW produces when it infests different host plants can provide a way to learn how the insect picks a specific type of plant [48]. This knowledge can be used to prevent future infestations of RPW in new host plants. In addition, characterizing the digestive enzymes in the RPW gut offers an opportunity to be used in designing promising inhibitors and establishing effective pest management strategies [50]. However, some of the enzymes have only been biochemically characterized and have not been tested with any inhibitor or insecticide yet. The proposed inhibitor or insecticide, however, needs to be designed as species-specific to avoid a leaching effect on nontarget organisms [100]. The symbiont-mediated RNAi (SMR) approach can be utilized to produce inhibitors for precise and specific pest management. This is an approach where symbiotic bacteria that have a limited host range were used to deliver insecticidal RNAi. This approach has been proposed to enhance the specificity of RNAi [101] and also been applied in Western flower thrips but not yet studied in RPW.

As part of RPW’s digestive system is lined with chitin, targeting the chitin biosynthesis pathway is also an opportunity for controlling and managing the pest [18]. Chlorfluazuron, a chitin synthesis inhibitor reported to have a low and negligible leaching rate (<7%) [102], was proven to be effective on RPW [18]. However, chitin is one of the basic components that build the exoskeleton in the most insects. Therefore, the usage of a chitin inhibitor may affect not only RPW but also other beneficial insects such as bees [103]. However, this specificity towards nontarget organisms can be tackled by utilizing a docking analysis between several 3D structures of chitinase from the potentially affected nontarget organisms and the inhibitor. This analysis could be achieved using advanced computational approaches in combination with emerging artificial intelligence (AI) technologies, such as AlphaFold [104,105] and AutodockVina [106,107]. Hence, the structures of potentially affected nontarget organisms are urgently needed [103].

## 8. Conclusions

The review showed that there are many opportunities to explore and study the digestive system of RPW as a means for managing and controlling the infestation and spread of RPW as an insect pest. Metagenomic, proteomic and transcriptomic data were shown to be suitable approaches to further understanding the protein and gene constituents of RPW. However, with the advancement of technologies, the detection limit, the data generated and the tools for analysis no longer limit deeper exploration at the systemic level, opening the doors for gaining more knowledge on the digestive system of RPW. Hence, a more holistic approach could be used to produce a better way to manage RPW.

## Figures and Tables

**Figure 1 insects-14-00506-f001:**
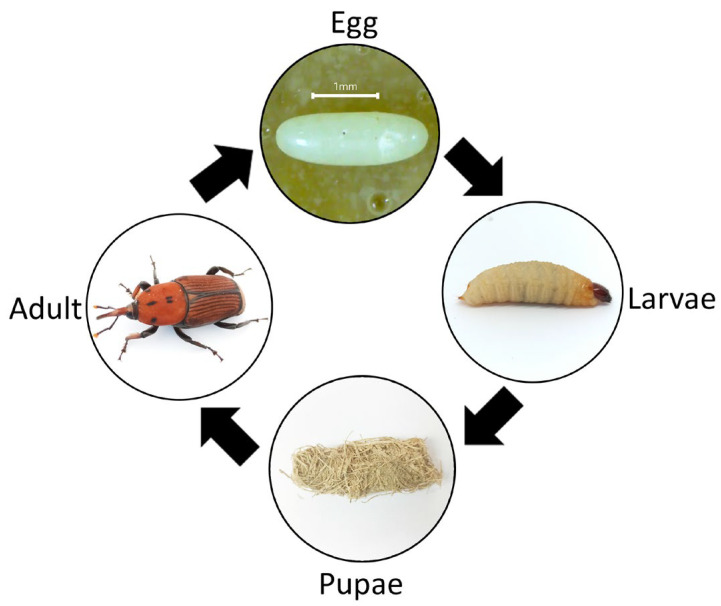
Lifecycle of RPW as holometabolous insects.

**Figure 2 insects-14-00506-f002:**
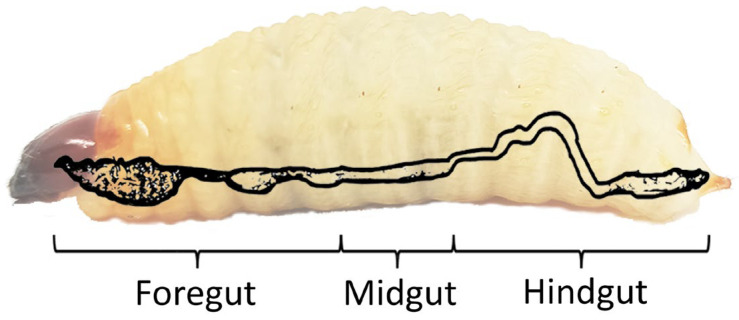
Internal digestive organ of an adult red palm weevil, *Rhynchophorus ferrugineus*. The gut is divided into three parts: foregut, midgut and hindgut.

**Table 1 insects-14-00506-t001:** Summary of potentially targeted enzymes found in the gut of RPW and their functions.

Enzyme	EC Number	Function
Trypsin	3.4.21.4	Protein digestion
Aminopeptidase	3.4.11.1	Protein digestion
Xylanase	3.2.1.8	Xylan (plant cell wall) digestion
Glycosidase	3.2.1	Hydrolyze polysaccharide of cell wall
Amylase	3.2.1.1	Starch digestion
Catalase	1.11.1.6	Catalyze conversion of hydrogen peroxide into oxygen and water
Peroxidase	1.11.1.7	Catalyze conversion of hydrogen peroxide into oxygen and water
Polyphenol oxidase	1.14.18.1	Catalyze the oxidation of phenolic compound using oxygen

**Table 2 insects-14-00506-t002:** Summary of potentially targeted genes/transcripts found in the gut of RPW and their functions.

Genes/Transcripts	EC Number	Function
Cytochrome P450	*	Catalyze oxidation of xenobiotics
glutathione-S-transferase	2.5.1.18	Added glutathione to oxidized xenobiotics
Laccase	1.10.3.2	Cuticle hardening
Carboxypeptidase	3.4.16.2	Protein digestion
Chitin synthase	2.4.1.16	Catalyzes the polymerization of chitin polymer
Cellulase	3.2.1.4	Cell wall degradation (cellulose)
Hemicellulase	3.2.1	Cell wall degradation (hemicellulose)
Pectinase	3.2.1.15	Cell wall degradation (pectin)
Neuropeptides	-	Trigger physiological process (digestion)

* Depends on the type of electron donor with which they interact [80]. - equals not available.

**Table 3 insects-14-00506-t003:** Summary of tested insecticides and their potential targets in RPW’s gut.

Insecticide/Inhibitor	Target Gene/Protein	Function of Target	Reference
*Thymus vulgaris* and *Ocimum basilicum* extract; soybean trypsin inhibitor; *N*-tosyl-l- phenylalanine chloromethyl ketone (chymotrypsin inhibitor)	Trypsin-like serine proteinase assessment; trypsin; chymotrypsin	Protein digestion	[50,51]
Hematoporphyrin dihydrochloride (photosensitizer)	Antioxidant enzymes (polyphenol oxidase; peroxidase)	Defense mechanism	[94]
Eserine (carbamate inhibitor)	Acetylcholinesterase (AChE)	Detoxifying enzyme	[16]
Protease inhibitor from palm dates kernel	Protease	Protein digestion	[95]
Novaluron	Chitinase	Chitin regulation	[18]
Sesquiterpene (Farnesol, Farnesyl acetate, Picrotoxin); Spinosad	Glutathione S transferase (GST), Cytochrome P450	Detoxification of xenobiotics	[77,96]
Spinosad	Nicotinic acetylcholine receptor and/or gamma Aminobutyric acid (GABA) receptor	Modulation of feeding behavior and reproduction	[96]
*Juniperus communis* essential oil	Gut protein content	Digestion system	[97]
RNAi/double strand RNA	Catalase	Defense mechanism	[61,66]
Aprotinin	Gut serine proteinase	Serine digestion	[98]
Protease inhibitor from *Vigna radiata* L. seeds	α-amylase	Carbohydrate digestion	[50,99]
Not tested	Neuropeptide precursor and receptor	To regulate physiology and behavior of insects	[90]
Not tested	Laccase	Oxidize toxic compounds ingested by the insect	[81]
Not tested	Xylanase	Digestion of plant cell wall	[53]
Not tested	Aminopeptidase	Protein digestion	[20]
Not tested	Cellulase	Digestion of plant cell wall	[88]
Not tested	Pectinase	Digestion of plant cell walls	[88,92]
Not tested	Glucosidase	Carbohydrate digestion	[56]

## Data Availability

No new data was created or analyzed in this study. Data sharing is not applicable to this article.

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
