# Peer review of "A Review on Digestive System of *Rhynchophorus ferrugineus* as Potential Target to Develop Control Strategies"

_insects, 2023, doi:10.3390/insects14060506_

Round 1
Reviewer 1 Report
A brief summary
Red palm weevil (RPW, Rynchophorus ferrugineus) is an economically importance pest of palm globally. The authors of this manuscript put together a review to highlight the potential use of its digestive system for managing this pest. The authors reviewed research that were done on anatomy, microflora, transcriptomic analysis, and proteomic analysis in relation to the RPW digestive system. Overall, the manuscript provided readers with good review as depicted by its title. The authors provided great introduction to the review topic. In the body of manuscript, having subtopics helped readers understand the points that authors attempted to make.
There are some grammatical errors throughout the documents that makes it rather difficult to understand; this can be improved.
Specific comments:
As mentioned above, there are some grammatical errors that will need to be addressed. I am not listing them one by one as that will be too many. My suggestion would be to have an English native speaker (or non-native English speaker who is fluent in English) who is a subject matter expert in the field to assist with this part. Some repetitive verbiage should be avoided. There are places in the manuscript that rewording or rephrasing is needed since they did not read quite right.
Also, each subtopic had some type of data from the reviews. Could those be presented in table format? It will be easier to follow.
Below are some suggestions (in bold) below to improve the manuscripts although I am pretty sure I missed some.
2. Anatomy of RPW’s digestive system (page 3)
Line 87, 91
Mouth à should be “mouthpart”. Mouth is incorrect verbiage.
Line 88
It is different from, not different with.
Line 91
In addition to, not addition off
Line 101
According to Harris et al. [15] à remove “the study by”
Line 103
Any hindrance to, not hindrance on
3. Gut bacteria of RPW (pages 3-4)
Line 116
RPW gut was also reported … (note: RPW gut did not do the reporting)
Lines 147-154
Who did the study? Any reference? It is not clear from the sentences here à reword or rewrite the sentences. Beetles are plural à should use were, not was.
Lines 158-162
… metatranscriptome approach is proposed … à Who proposed? Are the authors proposing this idea? It is not clear here.
4. Protein profile of RPW’s gut (pages 4-6)
Lines 178-179
Need to rephrase the first sentence; the verbiage “there was a study …” implies that it was a different study than reference [20].
5. RPW’s gut transcriptome analysis (pages 6-7)
Lines 284-286
I would remove the verbiage “A published study …” and rephrase or reword. It is obvious the study was published since you referenced it in your manuscript.
6. R. ferrugineus control related to its digestive system (pages 7-8)
Line 300
Change to Zulkifli et al. [20], instead of “As claimed by [20] …”
Lines 307-310
Rephrase or reword
Lines 312-317
Rephrase or reword
8. Conclusion
Line 373
I would use the word explore instead of exploit.
See comments above
Author Response
Thank you for reading our manuscript and reviewing it. Please see the attachment for the revised manuscript and the point-by-point responses to each of the comments and suggestions.

Author Response

(The authors gave the same response as above.)

Reviewer 3 Report
Dear Authors,
I read carefully your submitted article/review Insects-2374945, and I consider it a nice and interesting study on the digestive system of one of most important pest , the red weevil of palms. However, there are some points that I suggest to consider. So that, please, look at the revised word file of the mns attached below. I suggest to follow my notes , in order to improve the mns and its presentation.

The English language shows some inappropriate words or unclear phrases/periods, somewhere in the text. I suggest a language revision of the text by a mother language lecturer.
Author Response

(The authors gave the same response as above.)

Round 2
Reviewer 3 Report
Dear Authors,
I have read carefully your cover letter with your replies to all my comments/suggestions in the previous review. I also read the new version provided, following all my notes: I am fully satisfied and consider the new text as suitable to be published in the present form (see the attached pdf ).
Sincerely,
